# The Effect of Rotary Friction Welding Conditions on the Microstructure and Mechanical Properties of Ti6Al4V Titanium Alloy Welds

**DOI:** 10.3390/ma16196492

**Published:** 2023-09-29

**Authors:** Matúš Gavalec, Igor Barenyi, Michal Krbata, Marcel Kohutiar, Sebastian Balos, Milan Pecanac

**Affiliations:** 1Faculty of Special Technology, Alexander Dubček University of Trenčín, 911 50 Trenčín, Slovakia; matus.gavalec@tnuni.sk (M.G.); igor.barenyi@tnuni.sk (I.B.); michal.krbata@tnuni.sk (M.K.); 2Faculty of Technical Sciences, University of Novi Sad, 21000 Novi Sad, Serbia; sebab@uns.ar.rs (S.B.); pecanac.milan@uns.ar.rs (M.P.)

**Keywords:** homogeneous weld, Ti6Al4V, rotary friction welding, microstructure

## Abstract

The main task that the article introduces is the experimental study of how the geometry of contact surfaces affects the quality and mechanical properties of a rotary friction weld (RFW), as well as the findings of whether the RFW technology is suitable for the titanium alloy Ti6Al4V. The experiments were carried out for specimens with a diameter of 10 mm and were performed at 900 RPM. Three types of geometry were proposed for the RFW process: flat on flat, flat on 37.5° and flat on 45°. Based on these results, the best tested flat geometry was selected from the perspective of quality and economic efficiency. The welded joints were subjected to microstructural analysis, tensile testing, microhardness testing, and fractography, as well as spectral analysis of the fracture surface and EDS map analysis of oxygen. The flat geometry of the contact surface resulted in the least saturation with interstitial elements from the atmosphere. Fracturing in the RFW zone led to a brittle fracture with a certain proportion of plastic deformation. A pure ductile fracture occurred in specimens fractured in the HAZ region, where the difference in UTS values compared to specimens fractured by a brittle fracture mechanism was not significant. The average UTS value was 478 MPa.

## 1. Introduction

Rotary friction welding (RFW) is a solid-state joining process that has shown noticeable potential for the welding of dissimilar materials [1,2,3,4]. This process is predominantly used to weld cylindrical workpieces. One of the two cylindrical workpieces is kept stationary, while another workpiece is made to be rotated during the process. These workpieces are brought into mutual contact, which leads to intense frictional heating and severe plastic deformation at the contact surface area. The increase in temperature improves the diffusion flux between the materials in contact surface areas, resulting in strong metallurgical bonding at the interface [5,6]. RFW is a solid-state welding process that is becoming increasingly popular due to its advantages, such as a low weld heat input, a decrease in cost consumption, the simplicity of the device, and its environmental friendliness. RFW can be used to weld materials that are difficult to fuse, using fusion welding techniques [7,8]. RFW is the most popular technique for friction welding. Inertia friction welding and continuous drive friction welding represent two methods that may be used [9]. However, the RFW technique has a constraint: it cannot be used to weld components with a cross-section that is not circular [10,11,12]. According to reports, several welding factors, including the friction duration, forging time, friction pressure, forging pressure, and rotation speed [13,14,15], have an impact on the quality of friction welds. RFW typically entails two stages: a friction phase to provide the required heat and a forging phase to make the weld solidify [16]. As shown in the Akbarimousavi study [17], there is a direct connection between individual parameters in RFW. In the articles [17,18,19], a wide range of RPMs and corresponding friction pressures and forging pressures were used. This knowledge allows, by choosing different but corresponding parameters, the use of devices with different layouts. Based on the analysis of the microstructure, the saturation of the weld joint with interstitial elements from the atmosphere was monitored and the optimal geometry of the contact surfaces was validated from this knowledge. A static tensile test and fractographic analysis were performed on the welds with the optimal tested geometry to verify the quality of the RFW joint. Spectral analysis and a microhardness test were performed to confirm the hypotheses related to diffusion mechanisms. Historically, RFW was one of the first methods to have used friction as a solid-state welding technique. Meisnar et al. [4], in their research, provide concrete insights into how the consistent optimization of welding processes results in consistent and reliable welds with excellent mechanical properties. The aim of this work was to investigate the weld interface and microstructure of Ti6Al4V and AA6082 rotary friction weld joints for the purpose of spacecraft applications. The material Ti6Al4V is widely used for these applications, and therefore it is necessary to look for progressive joining methods. Jin et al. [20], in their work, focus on the morphology of the alpha phase in the titanium alloy, which significantly positively affects the strength in the area of the weld interface. Deformation of the alpha phase grains increases the strength in the weld interface area. This outcome is also confirmed for homogeneous RFW joints in this article.

Understanding internal material processes was the main goal of the investigation and it includes findings related to the crystal lattice of the given material and the physical mechanisms associated with the formation of weld joints [21]. Based on the knowledge obtained, which deals exclusively with the material aspects, the mechanism of RFW joint formation of the Ti6Al4V titanium alloy with the optimal selected geometry of the contact surfaces was formulated and described. Currently, it is possible to effectively create homogeneous and heterogeneous welds of titanium alloys using welding technologies, such as tungsten inert gas (TIG) welding [22] or, as Tomashchuk [23,24] and Wang [25,26,27] show in their studies, electron beam welding. Zhu [28] points out in his study that it is also possible to effectively use the CMT method to create joints of homogeneous titanium welds with regard to the microstructure. However, as Kundu [29] points out in his studies, it is also possible to use less common methods, such as diffusion welding.

## 2. Materials and Methods

During predetermined experiments, a manufactured Ti6Al4V (titanium alloy) was used in the shape of a round bar (diameter: 10 mm). The chemical composition of the base material TiAl4V in the delivered state is given in Table 1. In this basic state, the material reached an ultimate tensile strength (UTS) of 887 MPa and hardness of 340 HV10.

Ti6Al4V alloy has widespread industrial use due to its excellent physical and mechanical properties, which can be strengthened either by thermomechanical or heat treatment processes. Ti6Al4V alloy belongs to the dual-phase category of titanium alloys, with a microstructure comprising α- and β-phases [31,32]. The high cost of titanium alloys caused by the higher extraction costs and the complex nature of the secondary finishing operations (forming and machining) pose a severe challenge to their continued use in industrial applications [33,34,35].

The base material was in the form of round cold rolled bars, which were obtained after heat treatment. The standard heat treatment of the titanium alloy Ti6Al4V consists of solution treatment at a temperature of 940 to 970 ° C for 10 min with a water quench. The next step is ageing at a temperature of 480 to 595 ° C for 2 to 8 h with air cooling [36]. The structure of the base material after etching with 5 ml of HF + 20 ml of HNO_3_ + 85 ml of H_2_O is shown in Figure 1.

As mentioned above, Ti occurs in two allotropic modifications. Up to a temperature of 888 °C, there is the stable α-phase, which crystallizes in a tight hexagonal structure. Above this temperature, up to the melting temperature (1668 °C), there is the stable β-phase, which has a face-centered cubic structure. Titanium has a high affinity for oxygen; titanium oxide (TiO2) is quickly formed on the surface of the metal even at room temperature. However, since the molecular volume of the oxide is the same as the molecular volume of the metal, the oxide adheres to the surface, passivating it and protecting it from corrosion in salts or oxidizing acids and, to a greater extent, in mineral acids [37,38]. It is a two-phase titanium alloy, where aluminum acts as an α-phase stabilizer and vanadium as a β-phase stabilizer, and this makes this alloy two-phase even at room temperature. It is the presence of the β-phase that improves the weldability of the alloy and softens the grain, and thus it has a beneficial effect on the strength and toughness [39].

It is proven that hydrogen and oxygen are the main causes of pores [40]. During the cooling process of the weld metal, the solubility of hydrogen changes. For example, if the partial pressure of hydrogen in the atmosphere around the welding zone is high, the hydrogen in the weld metal cannot easily disperse and escape but collects and forms pores [41]. Moisture during welding and the presence of hydrogen also manifest in the porosity of the weld [37].

### Methodology

In this study, the RFW was performed on a SV 18 Ra TOS Trenčín lathe machine, which is adequate for the production of friction welds. As a fixation element in the non-rotating workpiece, the jig was used, and it was fixed in the tailstock of the lathe. The pressure of the workpiece was ensured by the extension of the tailstock spindle [21,41].

The pressure parameter was, in all experiments, chosen as the maximum amount of material resistance that the tail stock spindle allowed to develop. The relative movement speed was set at 900 RPM, and time was only a secondary parameter.

The reason for choosing three different geometries of the tested contact surfaces was related to the fact that each individual geometry showed a different degree of plastic deformation in the initial stages of the welding process, which led to different strengthening processes (cold forming and hot forming). The time for which the specimen was subjected to the strengthening process was very short and, therefore, the resulting structure was fine-grained, i.e., there was not sufficient time for grain coarsening to occur. An overview of the individual geometries tested is shown in Figure 2, where angle α = 45 ° and β = 37.5°.

In addition to line faults, i.e., dislocations, point defects arising on the contact surfaces in the initial stages of the welding process contribute in an invaluable way to the formation of a welded joint. In the initial stages, during the flattening of the microuniformities of the contact surfaces, as well as intensifying friction and increasing temperature, the deformation energy, supplied to the material in this way, generates a higher concentration of vacancies on the contact surfaces. Due to this higher concentration on the contact surfaces, diffusion processes between the contact surfaces take place actively at elevated temperatures and this facilitates the formation of the weld joint. It is related to the concentration balancing of atoms, which is described by Fick’s first law [21,41].

For microstructural analysis, conventional metallographic preparation was performed: cutting with emulsion cooling, grinding with SiC abrasive papers, and polishing with diamond suspensions. Etching was carried out with an etchant with a composition of 5 mL HF + 20 mL HNO_3_ + 85 mL H_2_O. The device for microstructural analysis was an NEOPHOT 32 optical microscope from the Carl Zeiss Company, Jena, Germany (Faculty of Special Technology, Slovakia). The static tensile tests for the specimens were performed using the Instron 5500R device (Faculty of Special Technology, Slovakia). The test methodology was according to the STN EN ISO 6892-1 standard [42]. After tensile testing, visual inspection was performed, and the electron microscopy evaluation of the fracture surfaces was performed using a TESCAN VEGA 3 scanning electron microscope (SEM) (Faculty of Industrial Technologies, Slovakia) operating at 30 kV. In addition, EDS analysis of selected elements on the surfaces of the quarries was also carried out [43,44].

Investigating the temperature intervals could help to better understand the process and extent of saturation with interstitial elements from the atmosphere for individual geometries of the contact surface. However, the measurement of the temperature development from the beginning of the welding cycle to the last stage of the RFW process could not be carried out due to the fact that standard instrumentation (infrared thermometers) could not accurately measure the temperature of the specimens, even after they were painted with matte paint. Moreover, the use of a thermocouple was not considered, due to the rotation of the moving specimen and clamping of the immovable in the support jig. The pressure required in the RFW process is seen rather as the maximum amount of resistance that the tailstock spindle of the lathe allows. Shortly before the end of the rotation, the material, under the influence of the high friction temperature of both welded parts, developed almost no resistance, and thus the pressure was minimal. Because the experimental measurements were performed on a lathe, the control process and the conservation of the same pressure force parameters on the tailstock spindle scale were applied. Using this value, the same length of material supply was ensured for each geometry tested to ensure the best possible weld uniformity.

## 3. Results

### 3.1. Macrostructure and Microstructure of the RFW Joints

From a macroscopic point of view, the friction weld was observed in all three cases, referring to the contact surface geometries. Relating to the first geometry, a detail of the macrostructure of the weld joint can be seen in Figure 3.

As one can see in Figure 3, this weld joint was essentially free of visible defects, and the color of the weld joint also revealed that the weld interface or heat-affected zone (HAZ) was not saturated with interstitial elements from the atmosphere. In addition, the given weld showed a blue coloration on the surface of the weldment in the HAZ, but there was not any effect on the integrity of the weld joint.

Images representing the detail of the microstructure of the mentioned weld joint are shown in Figure 4.

Figure 4 shows the weld interface, where the influence of the RFW process affected the structure of the base material after its heat treatment (solution treatment + ageing). The grain was formed into bands (β-phase) in the area of the weld interface. The creation of these bands clearly proves that the process of rotary friction welding and the forces acting on the material during the rotation of one of the welded parts in a certain way affect the direction of diffusion and thus also the final microstructure of the welded joint. Moreover, the gradual refinement of the grain, in the direction from the base material to the HAZ and the weld interface (where it is the finest), can be observed.

Based on the type of selected and used etchant, the β-phase (brown) was highlighted and merged, and its abundance was shown primarily in the weld area. This observation is in accordance with the literature, which states that there is a lack of α-phase in the Ti6Al4V alloy in the weld area. A higher temperature and a longer exposure time can lead to the greater conversion of the α-phase because a higher temperature can cause greater diffusion of Al and V (mainly Al diffusion) into the α-phase [45,46].

Figure 5 represents the case of the second geometry relating to the contact surfaces (flat surface—inner cone), where the saturation of the weld area and HAZ with interstitial elements from the atmosphere, as well as pores, could be clearly observed in the weld interface. Figure 5a shows an area in the middle of the weld, where a thicker weld area is clearly visible, while it slightly decreases towards the surface (Figure 5b).

In this case, the geometry of the contact surfaces played a crucial role in saturating the weld with interstitial elements: the hydrogen, oxygen, and nitrogen in the atmosphere were closed in the area of the internal cone during the rotary friction welding process, and the cavity was closed with the flat geometry of the second welded part. Based on this, it should be noted that there were excellent conditions for the diffusion of these interstitial elements into the titanium solid solution due to the appropriate local melting temperature.

In Figure 6, it can be seen that in addition to the dark β-phase, there was also a light α-phase in the solid solution. There are also continuous layers/dark bands in the given image, which are typical for the given area in the RFW process [47].

Based on the results for the second and the third geometries (inner cone—outer cone) of the contact surface, the effect was similar, as can be seen in Figure 7. There were suitable conditions created for interstitial diffusion into the area of the weld joint. In relation to the third geometry, diffusion did not take place primarily from the inside part of the weld joint (as in the case of the second geometry), but the process of diffusion occurred from the outside part of the specimen. Therefore, the area of interstitial saturation was smaller in comparison with the second geometry, because the diffusion took place against the movement of the centrifugal force. However, it is important to note that the interstitial diffusion into the weld interface took place almost during the entire welding process and thus during the overall time interval of the plasticized material’s spread. All processes proceeded in the direction from the axis of the welded parts outwards to the surface and subsequently to the flash, which was created by this plasticized material.

Figure 8 confirms the assumption that the third geometry of the contact surfaces contained a smaller number of layers/dark bands compared to the second geometry and was also not suitable for the production of Ti6Al4V titanium alloy weld joints.

### 3.2. Tensile Test

Based on the analysis of the microstructure, the static tensile test, fractography analysis, and microhardness analysis were performed only for the specimens referring to the first geometry of the contact surfaces (flat—flat), because the saturation with interstitial elements from the atmosphere interfered with the integrity of the rotary friction welds (the second and the third geometry).

The specimens for the static tensile test were not produced according to a standard because we did not want to weaken the weld area by removing the material from the friction weld flash and thereby introduce machining stress into the specimen or thermally affect the structure that we had already mapped based on the microstructure analysis. Thus, the specimens were subjected to a static tensile test in the design, as they were after welding.

All samples used for the static tensile test were created with a flat geometry of the contact surfaces. The results from the static tensile test can be found in Figure 9.

Individual diagrams made during the static tensile test are shown in Figure 10. The selected diagram for sample no. 7 (Figure 10a) corresponds to the diagrams in which the ductile fracture mechanism in the HAZ was confirmed. On the second diagram of the static tensile test for sample no. 5, which is shown in Figure 10b, a brittle fracture mechanism occurred and was confirmed in the weld interface region.

### 3.3. Fractography

In the case of the specimens designated as no. 7 and 8, fractures occurred outside the welding interface in the HAZ. In relation to these specimens, the fracture was undoubtedly based on a ductile fracture mechanism. These two values are very similar to the UTS values obtained by other researchers in their experiments [48]. Moreover, the given values correspond to approximately 60% of the UTS, guaranteed by the manufacturer. The average UTS value corresponds to approximately 54% of the UTS, guaranteed by the manufacturer.

Figure 11 represents the fracture surface of the specimen designated as no. 1 (Figure 9) and, in this case, the dominant brittle fracture mechanism with a certain proportion of plastic deformation on the perimeter was determined and observed for 1–6 specimens (Figure 9) after they had been subjected to a static tensile test. Some other researchers have also reported a predominantly brittle fracture mechanism in the weld area during the welding of this titanium alloy type [42].

The previous finding was also confirmed by the image of the brittle intercrystalline fracture surface from the SEM (Figure 12), which was obtained from the central part of the specimen, marked with 1 in Figure 11.

Regarding the spectral analysis of the fracture surface in Figure 13, the observation confirmed the different amounts of Ti on the individual fracture surfaces.

In Figure 14, we present the spectra related to the different morphological crack areas of the interface. In line with our predictions, it is indeed confirmed that the diffusion of interstitial oxygen from the atmosphere is most observed in the surface regions of the friction weld.

The results of the EDS analysis suggest that, during friction welding, localized changes in chemical composition occur, in terms of the moderate depletion of aluminum and vanadium in the weld area. This occurs due to the extreme heat and mechanical forces involved in the RFW process, which occur away from the weld area.

### 3.4. Microhardness

The microhardness test was performed with the Microhardness QATM Qness 10 (Faculty of Special Technology, Slovakia) device with a test load of 100 g.

Based on the microhardness test, the highest hardness was confirmed in the area with the largest amount of Ti in the matrix, i.e., in the weld interface area. This fact was also confirmed by the hardness values shown in the graph (Figure 15), where we can see that the highest hardness values were reached for the weld interface area and gradually decreased toward the HAZ on both sides. The line of indentions was guided along the axial axis of the weld, 4 mm on each side from the weld interface.

## 4. Discussion

A solid weld joint was created, from the microstructural point of view, only using the first geometry of the contact surfaces with the RFW method. In this case, no serious degradation from interstitial elements found in the atmosphere occurred. As a suitable indicator of the weld’s saturation with interstitial elements, in addition to the detailed analysis of the microstructure, the monitoring of the weld color spectrum was performed. A great advantage of this geometry is also the fact that there is no need for the laborious preparation of the contact surfaces before the actual welding process, and there is also no need for supervision to remelt the entire volume of the specially prepared geometry of the contact surfaces, as is the case with geometries two and three.

The closer it is to the circumference of the welded part, the greater the saturation of interstitial elements from the atmosphere. This is due to the suitable diffusion conditions of the interstitial elements of the atmosphere. Despite this, visible interstitial saturation, based on the color of the weld (weld cut), did not occur. The observed oxygen saturation, in the case of this geometry of the contact surfaces, was not large enough for any weldment to impair its integrity.

The findings based on our observations prove that when using the other tested geometries of contact surfaces (i.e., 2 and 3), the amount of distributed heat is greater than in the case of the first geometry, so, with these two geometries, the size of the thermal affect and the width of the HAZ are even greater, probably due to an increased contact area and deformation. Both of these contact surface geometries are not suitable for RFW of Ti6Al4V alloys.

The flat geometry of the contact surfaces does not allow interstitial elements (O, N, H) from the atmosphere to diffuse into the weld interface to a large extent. This cannot be said for the other two tested geometries, and this is also proven by the experimental results. Heating to the welding temperature in the RFW process takes approximately 20–22 s. Subsequently, after this time interval, there is almost no material resistance. The welding process itself takes another 6 to 7 s. The time required for the deep diffusion of interstitial elements from the atmosphere is not sufficient. It is also necessary to consider that the clamping of the weldment in the chuck on one side and in the clamping jig on the other side ensures high-quality heat removal from the welding equipment. Such conditions are far from guaranteed with conventional welding methods within the Ti6Al4V material [22].

A predominantly brittle fracture mechanism was observed, with a certain area of plastic deformation in the weld area. The results from the static tensile test confirmed the findings of other researchers [48]. In the case of the last two specimens, the material was fractured in the HAZ by a ductile fracture mechanism. The welding parameters and material were unchanged for these specimens. The difference in UTS between the specimens broken by the brittle fracture mechanism and the ductile fracture mechanism was very small (in half of the tested specimens), so the measured values can be considered as the borderline between brittle fracture and ductile fracture. It should be noted that the RFW technology does not require the use of any protective atmosphere. Conventional welding technologies, used during titanium alloy welding, require a massive amount of extremely clean protective atmosphere, which of course is reflected in the price of the weld. This technology offers, from an economic point of view, the unique possibility of welding titanium alloys with obvious resource (wire, gas, time) savings [22,25,28].

The RFW process refines the grain of the material in the direction from the weld interface (finest) to the base material (original—coarser) [18]. Under normal circumstances, a fine-grained structure would cause greater toughness, i.e., a larger amount of grain boundaries that form a natural obstacle to dislocations, which are formed during plastic deformation. However, the fact that we recorded a lower strength limit than the base material can be explained by the shape of the weldment after the RFW process and thus the shape of the friction weld flash, which can be the initiator of the fracture at the weld interface. Moreover, a larger amount of grain boundaries causes higher hardness in the weld joint area and a gradual decrease in hardness towards the base material.

In future experiments, it would be appropriate to focus on the effect of the friction weld flash on the value of the ultimate tensile strength. For relevant results, the process and parameters should be kept as described in the article.

## 5. Conclusions

The investigation of the RFW weld of the Ti6Al4V alloy presented in the article can be concluded as follows.

The most reliable geometry of the contact surfaces was proven to be geometry number one—the geometry without any special modification of the contact surfaces.With the mentioned geometry, there was the least saturation with interstitial elements of the atmosphere. The remaining two tested geometries of the contact surfaces were not suitable for the RFW of the titanium alloy Ti6Al4V, precisely due to their tendency to be saturated with interstitial elements from the atmosphere.The static tensile test confirmed the brittle fracture with a certain proportion of plastic deformation. These specimens were fractured in the RFW area. The ductile fracture mechanism was observed in two specimens, but the difference in the UTS value compared to the specimens broken by the brittle fracture mechanism was not significant. The specimens in which a ductile fracture mechanism was observed were fractured outside the weld interface, in the HAZ.The average value of the ultimate tensile strength reached approximately half of the UTS limit of the base material. On the basis of the experimental results, it was possible to determine the appropriate tested geometry of the contact surfaces and also to understand the effect of saturating the weld joint with interstitial elements from the atmosphere, in a process in which no protective atmosphere or filler material was used.

## Figures and Tables

**Figure 1 materials-16-06492-f001:**
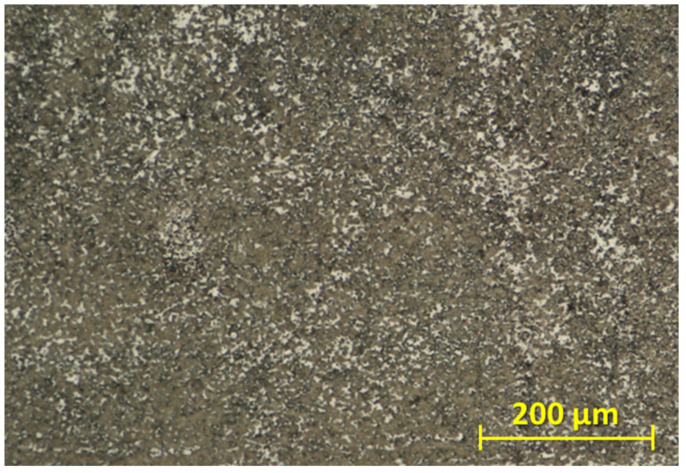
Microstructure of the base material after etching.

**Figure 2 materials-16-06492-f002:**
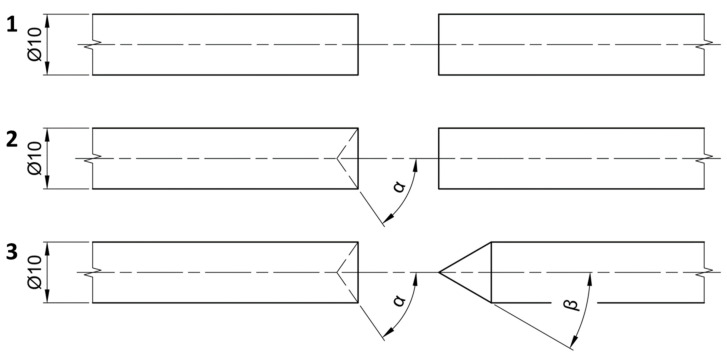
Overview of individual tested geometries.

**Figure 3 materials-16-06492-f003:**
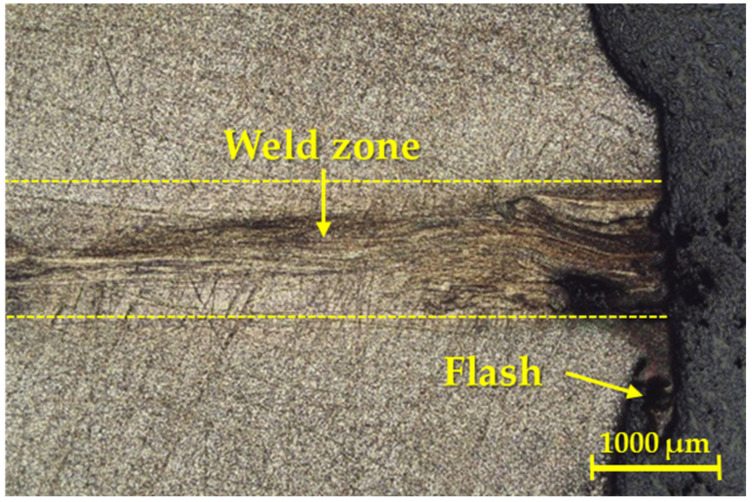
Detail of the macrostructure of the weld joint for the first geometry.

**Figure 4 materials-16-06492-f004:**
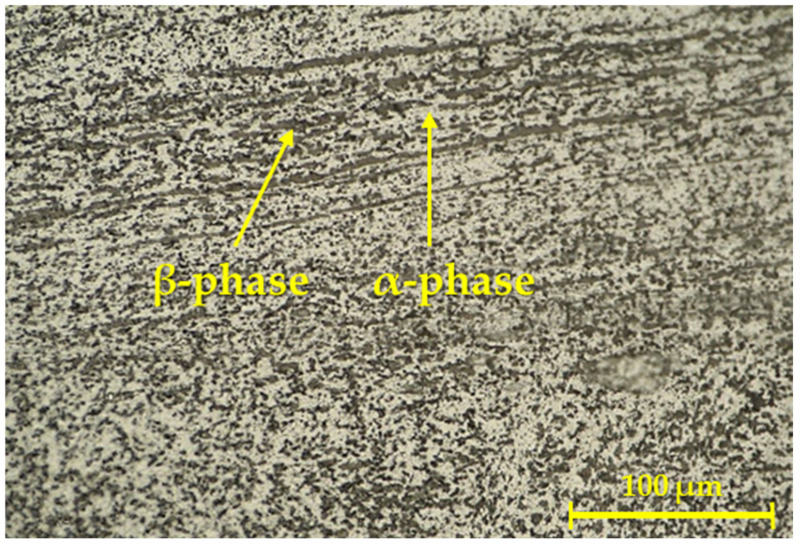
Microstructure of the weld joint for the first geometry.

**Figure 5 materials-16-06492-f005:**
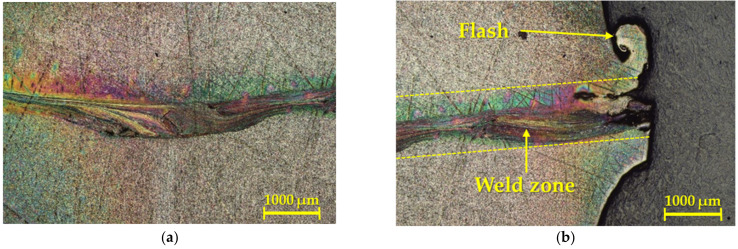
Detail of the macrostructure of the weld joint, created for the second geometry: (**a**) middle of the weld, (**b**) edge of the weld.

**Figure 6 materials-16-06492-f006:**
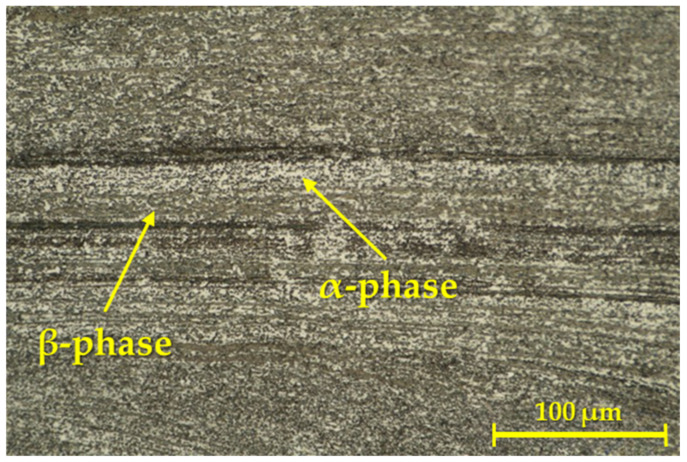
Microstructure of the weld joint, created for the second geometry.

**Figure 7 materials-16-06492-f007:**
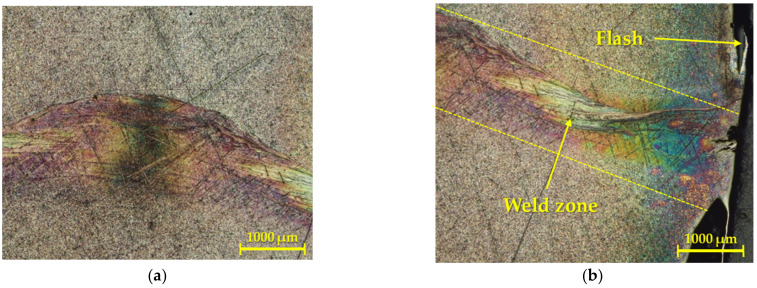
Detail of the macrostructure of the weld joint created for the third geometry: (**a**) middle of the weld, (**b**) edge of the weld.

**Figure 8 materials-16-06492-f008:**
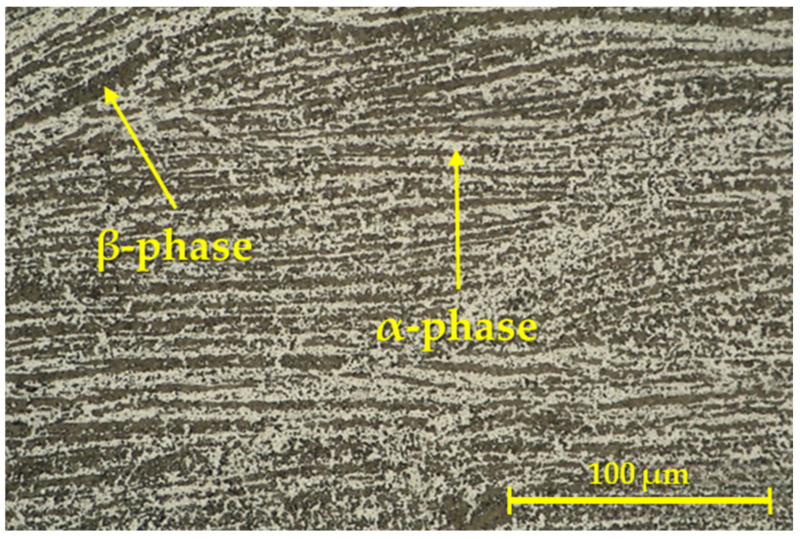
Microstructure of the weld joint created by the third geometry.

**Figure 9 materials-16-06492-f009:**
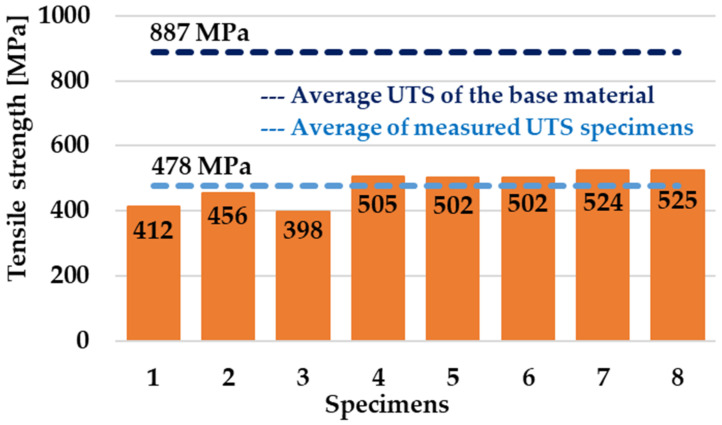
UTS of the experimental friction welds.

**Figure 10 materials-16-06492-f010:**
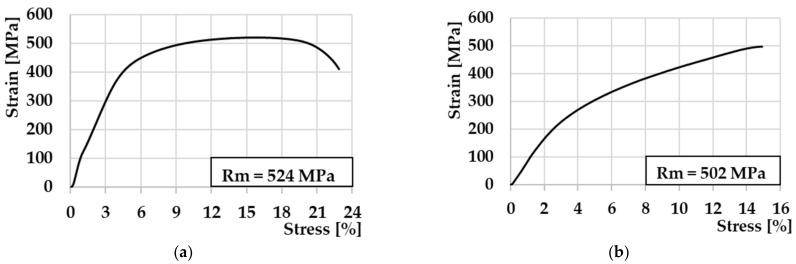
Selected diagrams from the static tensile test: (**a**) specimen no. 7—fracture in HAZ; (**b**) specimen no. 5—fracture in the middle of the weld joint.

**Figure 11 materials-16-06492-f011:**
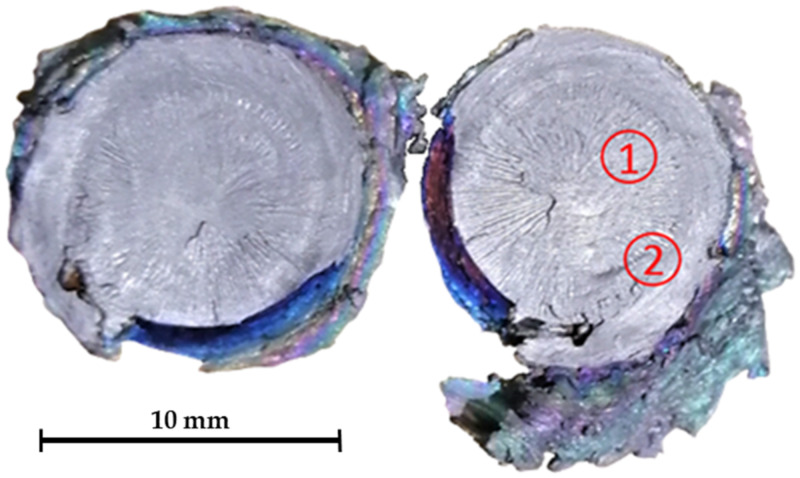
Fracture surface of the specimen after the static tensile test. 1. Brittle intercrystalline fracture zone, 2. Interface of brittle intercrystalline fracture zone and zone with increased plastic deformation.

**Figure 12 materials-16-06492-f012:**
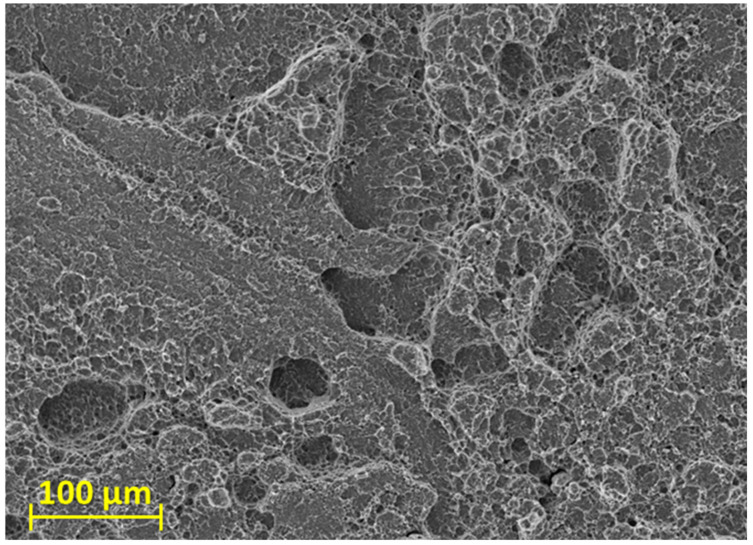
SEM image of brittle intercrystalline fracture observed in position 1.

**Figure 13 materials-16-06492-f013:**
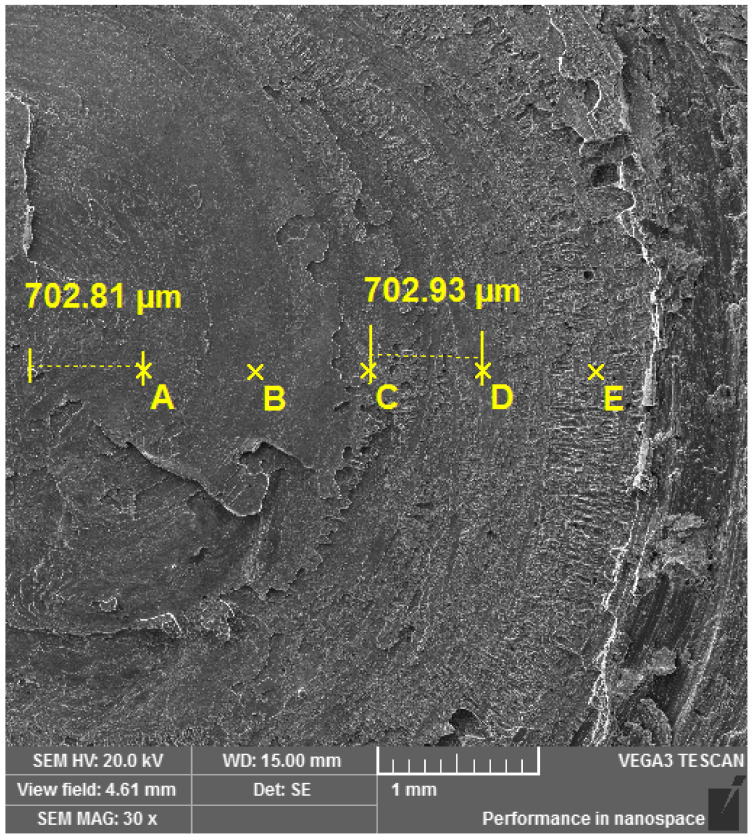
Locations (A–E) of EDS analysis.

**Figure 14 materials-16-06492-f014:**
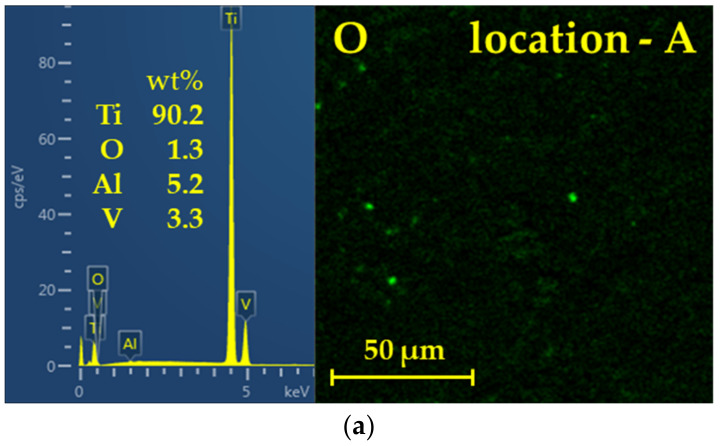
EDS analysis of the rotary friction weld fracture surface: (**a**) location A; (**b**) location B; (**c**) location C; (**d**) location D; (**e**) location E.

**Figure 15 materials-16-06492-f015:**
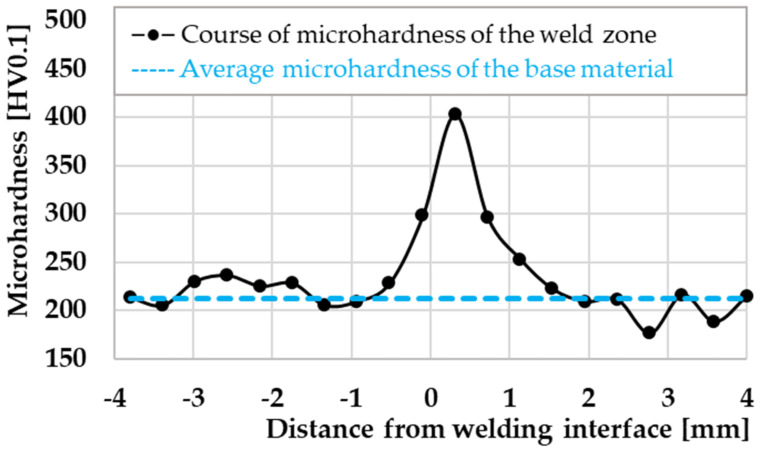
Course of microhardness across the weld zone.

**Table 1 materials-16-06492-t001:** Chemical composition of Ti6Al4V titanium alloy (wt.%) [30].

Element	Al	V	H	O	N	C	Fe	Ti
(wt.%)	5.95	4.00	0.00	0.20	0.01	0.03	0.04	Balance

## Data Availability

Data are available upon request to the corresponding author.

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
