# Peer review of "The Effect of Rotary Friction Welding Conditions on the Microstructure and Mechanical Properties of Ti6Al4V Titanium Alloy Welds"

_materials, 2023, doi:10.3390/ma16196492_

Round 1
Reviewer 1 Report
In this paper, the effect of RFW on the microstructure and mechanical properties of Ti6Al4V titanium alloy joint is discussed, and the influence of the choice of contact surface geometry on the welding quality is analyzed. This makes sense, but the following questions still need to be answered:
1) The introduction section briefly describes the purpose and method of the research, and suggests adding a more explicit statement of the research problem or innovation point to highlight the contribution and significance of the article. In addition, in view of the introduction of RFW, it is suggested to properly summarize and evaluate the existing research of RFW in titanium alloy welding, in order to show the connection and difference between this paper and previous work.
2) The experimental parameters and conditions are given in section 2.1, but there is no assessment of possible errors or uncertainties in the experiment. It is recommended that some content be added to describe the factors in the experiment that may affect the accuracy or reliability of the results, and what measures have been taken to reduce or eliminate these factors.
3) The third chapter describes the experimental phenomena and observation results of the three mechanical experiments. Is it possible to add some discussion and explanation in order to reveal the influence mechanism of RFW process on the mechanical properties of Ti6Al4V alloy, so as to make the experimental results more convincing?
4) The fourth chapter does not clearly put forward the limitations and shortcomings of this paper or the future research direction and prospect. It is suggested to add a description of the problems or room for improvement in experimental design, data processing, result analysis, etc., and put forward some possible solutions or suggestions. Or what questions and hypotheses need to be further explored or verified based on the research content of the article.
Reviewer 2 Report
Review report
on manuscript entitled “The Effect of Rotary Friction Welding Conditions on the Change of Microstructure & Mechanical Properties of Ti6Al4V Titanium Alloy Welds” by the authors Matus Gavalec, Igor Barenyi, Michal Krbata, Marcel Kohutiar, Sebastian Balos and Milan Pecanac (Manuscript ID: materials-2599508)
The focus of the present study is placed on the search for an optimal design of contact surface for rotary friction welding (RFW) of Ti-6Al-4V alloy. Based on experimental observations, the flat contact surface was deduced to be most suitable for this purpose. It was also concluded that the approach elaborated in the present work is feasible for sound welding of Ti-6Al-4V without using shielding gases.
I do believe that the topic of this manuscript represents an essential interest for the welding community. However, if the authors would like to publish their results in Materials (thus sharing them with the materials community), they perhaps should pay more attention to the underlying metallurgical phenomena, e.g., microstructural evolution, microstructure-property relationship, etc.
I disagree with the main conclusions derived by the authors from the experimental results. The strength of the welded joints was approximately twice lower than that of the base material (Fig. 8); moreover, the welded joint typically failed in the weld zone. In this context, of particular interest was the observation that the local hardness of the material within the weld zone was nearly twice higher than that of the base material (Fig. 13). Given the typically brittle character of the weld failure (Figure 9), the above facts suggest that the welded joints likely contained macro-scale defects. If so, the welding approach developed by the authors was far successful.
An alternative explanation for the inferior weld strength may be the extensive entrapment of oxygen by the welded material during the RFW process. This may provide material embrittlement and thus lead to a premature fracture. This suggestion is in line with an increased oxygen content in the failure location (Fig. 12). If so, the authors’ conclusion on the feasibility of welding of Ti-6Al-4V without shielding gases is also not correct.
In addition to the above principal comments, there are also several technical issues, which require consideration by the authors.
(1) In Introduction section, it would be useful to outline the current state-of-the-art in the field of the effect of RFW conditions (including rotation rate, imposed pressure, dwell time, shielding gases, etc.) on the microstructure and properties of welded joints.
(2) The term “interstices” seems strange. Please double-check it.
(3) In Experimental section, please provide a thermo-mechanical history of the base material and briefly describe its microstructure prior to the RFW process.
(4) In Experimental section, please provide all details of the RFW procedure, which are sufficient for the reproduction of this experiment by any independent researcher. Particularly, please indicate the applied pressure and dwell time.
(5) It would also be useful to indicate the design and dimensions of tensile specimens machined from the welded joints.
(6) In microstructural observations of Ti-6Al-4V with optical microscopy, the alpha-phase usually appears bright while the beta-phase is dark. So, please check the labels in the optical micrographs in Figs. 3, 5, and 7.
(7) In my opinion, there is no evident porosity or cracking in Figs. 4 and 6.
(8) It would be very useful to show deformation diagrams recorded during the tensile tests. Also, it would be great to show the appearance of the failed tensile specimens.
(9) It is very strange that the EDS analysis of the failed specimens (Fig. 12) revealed no presence of aluminum and vanadium! Please double-check this issue.

Reviewer 3 Report
This manuscript studies the effect of contact surface geometry on the mechanical properties of the Ti6Al4V rotary friction welds (RFW) specimens. The article needs the following corrections, which should be addressed.
The title of the article can be shorter. It is suggested to delete the term “changes of”.
The abstract can be made more attractive by using more quantitative data. It is suggested to provide more quantitative results.
The novelty and purpose of the research should be clearly stated in the abstract and introduction.
The introduction is brief, superficial, and incomplete. The number of used and reviewed references is minimal. Also, the paragraphs presented are primarily general and general information. At the end of the introduction, a suitable summary of the importance of the present issue should be provided.
Use the following resources to deepen the introduction. Effect of heat input on interfacial characterization of the butter joint of hot-rolling CP-Ti/Q235 bimetallic sheets by Laser + CMT. Hydrogen embrittlement behavior of SUS301L-MT stainless steel laser-arc hybrid welded joint localized zones.
The material and method section, as well as its sub-section, is very long and tedious. You can use the table and schematic to categorize and summarize this section.
How have the welding quality and reproducibility been checked?
The SEM images are raw, and labels, scale bars, and descriptions should be added. Also, the quality of these images is not very good.
Why is the stress-strain diagram not provided? To check and compare the mechanical properties, it is necessary to present the stress-strain diagram and the quantities of elastic modulus, yield stress, ultimate tensile stress, and ultimate and failure elongation.
The sources used must be up-to-date. Also, the number of sources in the discussion section is very limited.
***
Round 2
Reviewer 2 Report
Review report
on manuscript entitled “The Effect of Rotary Friction Welding Conditions on the Microstructure & Mechanical Properties of Ti6Al4V Titanium Alloy Welds” by the authors Matus Gavalec, Igor Barenyi, Michal Krbata, Marcel Kohutiar, Sebastian Balos and Milan Pecanac (Manuscript ID: materials-2599508)
This article is a revised version of the previously submitted manuscript. Unfortunately, I still cannot recommend it for publication because of the reasons outlined below.
(1) In my opinion, conclusion #4 is not correct. The strength of the welded joints was approximately twice lower than that of the base material (Fig. 9); moreover, 6 out of 8 welded joints failed in a brittle manner, with the failure occurring in the weld zone (Fig. 11). These results are in the clear contrast with a number of recent works in the scientific literature, which reported that the strength of the RFW-joints of Ti-6Al-4V is comparable with that of the base material. Several examples are given below.
(a) Feng Jin, Haodong Rao, Qian Wang, Guodong Wen, Pu Liu, Jiatao Liu, Junjun Shen, Jinglong Li, Jiangtao Xiong, Ninshu Ma, Heat-pattern induced non-uniform radial microstructure and properties of Ti-6Al-4V joint prepared by rotary friction welding, Materials Characterization, Volume 195, 2023, 112536, https://doi.org/10.1016/j.matchar.2022.112536.
(b) K. Sri Ram Vikas, K. Srinivasa Rao Rahul, G. Madhusudhan Reddy, and V.S.N. Venkata Ramana, Influence of hat treatments on microstructural and mechanical properties of Grade 5 titanium friction welds, Eng. Res. Express, Volume 4, 2022, 025053, https://iopscience.iop.org/article/10.1088/2631-8695/ac7a0a
This comparison indicates that the particular approach employed in the present study is perhaps not feasible for the production of high-quality RFW-joints of Ti-6Al-4V.
(2) EDX measurements revealed no presence of aluminum and vanadium in the weld zone (Fig. 14). In my opinion, this result is very strange and requires an additional experimental double-check.
(3) Given the relatively high hardness measured in the weld zone (Fig. 15), the depletion of the welded material by vanadium and aluminum (proposed by the authors) seems unlikely. It is hard to believe that the grain refinement in this microstructural region can overbalance the depletion effect.
(4) Please follow appropriate standards during the machining of tensile specimens and conducting the tensile tests. Otherwise, your experimental results are difficult to compare with the literature data. Considering the relatively high melting point of Ti-6Al-4V, the machining of the tensile specimens should introduce only negligible changes into the inner structure of the welded joints.
(5) Please double check the magnitude of the plastic strain indicated in the abscissa of the deformation diagrams in Fig. 10 (as well as that in Table 2). The ductility of 78.8 pct. (or even 39.2 pct.) is excessively large for Ti-6Al-4V.
(6) In microstructural observations of Ti-6Al-4V with optical microscopy, the alpha-phase usually appears bright while the beta-phase is dark. This idea agreed well with the examples provided by the authors in the response letter. Thus, I still recommend checking the labels in the optical micrographs in Figs. 4, 6, and 8.
(7) It would be great to see the appearance of the failed tensile specimens.

Round 3
Reviewer 2 Report
In my opinion, the revised manuscript is suitable for publication